# Phase I/II Trial of Urokinase Plasminogen Activator-Targeted Oncolytic Newcastle Disease Virus for Canine Intracranial Tumors

**DOI:** 10.3390/cancers16030564

**Published:** 2024-01-29

**Authors:** John H. Rossmeisl, Jamie N. King, John L. Robertson, James Weger-Lucarelli, Subbiah Elankumaran

**Affiliations:** 1Veterinary and Comparative Neuro-Oncology Laboratory, Department of Small Animal Clinical Sciences, Virginia-Maryland College of Veterinary Medicine, Virginia Tech, Blacksburg, VA 24061, USA; jnking@vt.edu (J.N.K.); drbob@vt.edu (J.L.R.); 2Department of Biomedical Sciences and Pathobiology, Virginia-Maryland College of Veterinary Medicine, Virginia Tech, Blacksburg, VA 24061, USA; weger@vt.edu (J.W.-L.)

**Keywords:** dog, brain, central nervous system, glioma, meningioma, paramyxovirus, viroimmunotherapy

## Abstract

**Simple Summary:**

Neurotropic oncolytic viruses have shown promise for the treatment of brain tumors, as they are naturally capable of entering the brain and selectively killing cancer cells. In this study, the safety, immunologic responses, and anti-tumor effects of intravenous administration of a genetically modified strain of neurotropic oncolytic Newcastle disease virus (NDV) to 20 dogs with spontaneously occurring brain cancers are characterized. Viral treatment was safe, with common side effects limited to transient low-grade fever, chills, and diarrhea. Anti-tumor responses, defined by a post-treatment reduction in tumor size as measured with brain MRI scans, were observed in two dogs. NDV genetic material was detectable in canine tumor tissue after treatment, confirming the ability of NDV to infect tumors. All dogs rapidly developed antibodies to NDV, suggesting that the viral dosing schedule may require modification to improve anti-tumor effects.

**Abstract:**

Neurotropic oncolytic viruses are appealing agents to treat brain tumors as they penetrate the blood–brain barrier and induce preferential cytolysis of neoplastic cells. The pathobiological similarities between human and canine brain tumors make immunocompetent dogs with naturally occurring tumors attractive models for the study of oncolytic virotherapies. In this dose-escalation/expansion study, an engineered Lasota NDV strain targeting the urokinase plasminogen activator system (rLAS-uPA) was administered by repetitive intravenous infusions to 20 dogs with intracranial tumors with the objectives of characterizing toxicities, immunologic responses, and neuroradiological anti-tumor effects of the virus for up to 6 months following treatment. Dose-limiting toxicities manifested as fever, hematologic, and neurological adverse events, and the maximum tolerated dose (MTD) of rLAS-uPA was 2 × 10^7^ pfu/mL. Mild adverse events, including transient infusion reactions, diarrhea, and fever were observed in 16/18 of dogs treated at or below MTD. No infectious virus was recoverable from body fluids. Neutralizing antibodies to rLAS-uPA were present in all dogs by 2 weeks post-treatment, and viral genetic material was detected in post-treatment tumors from six dogs. Tumor volumetric reductions occurred in 2/11 dogs receiving the MTD. Systemically administered rLAS-uPA NDV was safe and induced anti-tumor effects in canine brain tumors, although modifications to evade host anti-viral immunity are needed to optimize this novel therapy.

## 1. Introduction

As in humans, dogs naturally develop brain tumors representing a wide range of histologies including meningiomas, neuroepithelial neoplasms (astrocytoma, oligodendroglioma, ependymoma), pituitary tumors, and metastatic carcinomas [1,2,3]. Post mortem-based studies also indicate that brain tumors account for ~2–5% of all cancers, an incidence that approximates that seen in humans [1]. Clinical management of brain tumors in pet dogs also closely parallels approaches used in humans, with surgery, radiotherapy, and chemotherapy remaining important therapeutic modalities, which can improve the quality and quantity of life in patients with a variety of brain tumors [1,4,5]. However, as also occurs in humans, despite multimodality treatments, tumor recurrence and therapeutic resistance remain major modes of treatment failure and sources of morbidity and death, particularly with malignant brain tumors [4,5]. Thus, there are currently unmet clinical needs to develop and evaluate novel treatments that can further improve outcomes in humans and dogs with brain tumors. As dogs with spontaneously arising brain tumors are immunocompetent, have tumors with pathobiological similarities to their human counterparts, share their environment with people, and are treated with the same methods, they are an attractive model for comparative oncology studies [1,6].

Several intrinsic features of neurotropic oncolytic viruses (NOVs) are promising for the treatment of brain cancers. NOVs are capable of crossing the blood–brain barrier, inducing preferential cytolysis of rapidly dividing cancer cells throughout the brain and stimulating host production of cytokines with anti-tumor activity [7,8]. NOVs may also be further engineered for specific or enhanced anti-tumor activity through the targeting of tumor-associated proteins or through the direct production of immunostimulatory molecules [7]. Newcastle disease virus (NDV) is an enveloped, negative-sense, single-strand ribonucleic acid avian paramyxovirus, and is an attractive therapeutic NOV candidate as it was well tolerated and demonstrated a preliminary efficacy signal in humans with glioblastoma, and historically has a favorable safety profile when administered systemically to dogs as a vaccine vector [8,9,10].

In this study, a tropism-modified and protease-activated recombinant Lasota NDV strain with urokinase plasminogen activator modified fusion (F) protein cleavage site (rLAS-uPA) was administered by serial intravenous (IV) infusions to dogs with intracranial tumors. Proteolytic cleavage of the rLAS-uPA F protein, which is a major determinant of the cellular infectivity of NDV, occurs exclusively upon binding to the urokinase plasminogen activator receptor (uPAR), which has been shown to be overexpressed in several common histologic types of canine brain tumors [11,12]. It was hypothesized that systemically administered rLAS-uPA would be well tolerated, capable of infecting tumors, and able to induce objective anti-tumor effects in dogs with brain tumors. The study objectives were to: (1) identify the maximally tolerated dose (MTD) of and describe adverse events associated with systemic delivery of rLAS-uPA; (2) determine if treatment with MTD rLAS-uPA resulted in a quantitative anti-tumor response, defined by a reduction in tumor volume as determined with brain magnetic resonance imaging (MRI) examinations; and (3) characterize the immune responses to rLAS-uPA therapy in dogs with naturally occurring intracranial tumors. 

## 2. Materials and Methods

### 2.1. rLAS-uPA Viral Formulation and In Vitro Viral and Immunological Analytical Methods

#### 2.1.1. Formulation of rLAS-uPA Recombinant NDV

The formulation of the rLAS-uPA from the full-length Lasota NDV strain (GenBank Accession: AF077761) and confirmation of rLAS-uPA cytotoxicity on uPAR expressing U-87 tumor cells (U-87 MG; ATCC, Manassas VA, USA) were performed as described for the generation of the recombinant NDV with F protein cleavage site targeted to prostate-specific antigen [13], except that in this study, the F protein cleavage site was targeted to uPA. The rLAS-uPA was grown, rescued from complementary cDNA, amplified, and purified, stored, and titered using previously described methods [13,14].

#### 2.1.2. Cell Lines

Human epithelial carcinoma cells (HeLa CCL2; ATCC), for viral replication, transfection, and rescue methods, were grown and maintained in Dulbecco Modified Eagle Medium (DMEM; Thermo Fisher, Waltham, MA, USA) supplemented with 10% heat-inactivated fetal bovine serum (FBS; Atlanta Biologicals, Lawrenceville, GA, USA), 100 μg of penicillin/mL, and 0.1 μg of streptomycin/mL (Invitrogen, Carlsbad, CA, USA), in a 5% CO_2_ incubator at 37 °C. The DF-1 chicken embryo fibroblast cell line, which was utilized for viral rescue and virus neutralization assays, (UMNSAH/DF-1; ATCC) was maintained in DMEM supplemented with 10% FBS. For use in rLAS-uPA in vitro cytotoxicity studies, canine J3T and human U-87 (U-87 MG; ATCC) glioma cell lines were cultured in RPMI-1640 (Cellgro, Mediatech, Manassas, VA, USA) with supplemental 10% FBS in a 5% CO_2_ incubator at 37 °C. The canine J3T cell line was obtained from Dr. Michael Berens (Barrow Neurological Institute, Phoenix, AZ, USA) [15].

#### 2.1.3. Virus Neutralization Test

The virus neutralization test was carried out using pre- and post-treatment canine sera in DF-1 seeded in 96-well tissue culture plates. Two-fold serial dilutions of 50 μL of complement-inactivated canine serum samples were incubated for 1 h with 100 TCID_50_ rLAS-UPA in DMEM. Viral titers were measured by endpoint dilution assay, and the TCID_50_ was calculated as described by Reed and Muench [16].

#### 2.1.4. Cell Viability Assays and Syncytia Formation

Cell viability was quantified using MTT assays [3-(4,5-dimethylthiazol-2-yl)-2,5-d phenyltetrazolium bromide; Thermo Fisher]. Canine J3T and human U-87 glioma cells were seeded on 96-well plates (5000 cells/well), and then the cells were PBS (Thermo Fisher) vehicle-infected or infected with 0.01, 0.1, 1, and 10 MOI of rLAS-uPA expressing enhanced green fluorescent protein (rLAS-uPA-eGFP) for 24, 48, and 72 h. The rLAS-uPA-eGFP was formulated using previously described methods and has an extra cistron encoding eGFP inserted between the P and M gene sequences of the rLAS-uPA stain [8]. rLAS-uPA-eGFP and PBS vehicle-infected cells were examined for syncytia formation using microscopy. Absorbance was spectrophotometrically measured at 570 nm using a microplate reader (Spectramax Plus 384, Molecular Devices, San Jose, CA, USA) [17].

#### 2.1.5. Infectious Viral Recovery

Viral recovery was performed using previously described methods [14]. Sample supernatants (whole blood, CSF, or urine) were obtained through clarification by centrifugation at 1000× *g* for 10 min at 15 °C. Aliquots of 0.2 mL of supernatant were inoculated into the allantoic cavity of each of five embryonated SPF chicken eggs (Charles River Laboratories, Norwich, CT, USA) of 9–11 days incubation. Following inoculation, eggs were incubated at 37 °C for 4–7 days. Checks for bacterial contamination were performed by streaking samples in Luria Broth agar plates and reading these at 24 and 48 h of incubation against a light source. Contaminated samples were treated by incubation with increased antimicrobial concentrations for 2–4 h (gentamicin, penicillin G, and amphotericin B solutions [Invitrogen] at maximum final concentrations of 1 mg/mL, 10,000 U/mL, and 20 µg/mL, respectively). To accelerate the final isolation, two passages were performed at the 3-day interval, obtaining results comparable to two passages at 4–7-day intervals. Eggs containing dead or dying embryos, and all eggs remaining at the end of the incubation period, were chilled to 4 °C for 4 h or overnight and the allantoic fluids were tested for hemagglutination activity [14]. Fluids that yielded a negative reaction were passed into two further batches of eggs. 

#### 2.1.6. rLAS-uPA qRT-PCR in Brain Tumor Samples

Quantitative RT-PCR for rLAS-uPA was performed using the method described by Seal et al. [18]. Brain tumor samples (three 5 mg biological replicates from each dog) were used to extract mRNA using an RNeasy kit (Qiagen, Valencia, CA, USA), according to the manufacturer’s instructions. The primers used for RT-PCR (sense-5′-TCGAGICTGTACAATCTTGC-3′ and antisense-5′-GTCCGAGCACATCACTGAGC-3′) were generated by alignment of published NDV nucleotide sequences (GenBank Accession: AAC55035) representing a segment of the open reading frame for the matrix protein using Primer Express software (v.3.0.1; Applied Biosystems, Foster City, CA, USA), resulting in a 232 bp amplicon [18]. TaqMan PCR primers for the canine housekeeping gene glyceraldehydes-3-phosphate dehydrogenase (GAPDH) were used, and rLAS-uPA expression data were normalized to GADPH expression.

#### 2.1.7. Serum Cytokine Analyses

Commercially available, canine-specific ELISA kits were used to measure tumor necrosis factor-alpha (Canine TNF-α Quantikine ELISA, R&D Systems, Minneapolis, MN, USA), interferon alpha (Canine IFN-α ELISA, Invitrogen, Carlsbad, CA, USA), and TNF-related apoptosis-inducing ligand/Apo2L (Canine TRAIL ELISA, Invitrogen) cytokines in dog sera. Assays were performed in duplicate using 100 μL aliquots of pre- and post-immune canine sera serially diluted (1:10–1:100,000) in antibody-conjugate diluent according to the manufacturer’s instructions. Microplates were read at 450 nm using a spectrophotometer (Spectramax Plus 384, Molecular Devices). 

### 2.2. In Vivo Canine Clinical Trial Protocol and Clinical Endpoints

This was a prospective, single-center, open-label, Phase I/II basket trial for dogs with solitary intracranial tumors utilizing a 3 + 3 dose-escalation schema (Figure 1) [19]. Trial procedures were approved by the institutional animal care and use committee (protocol 15-103), and dog owners provided written, informed consent for their animal to receive treatment. For inclusion in both phases of the trial, dogs were required to have a Karnofsky Performance Score (KPS) ≥ 60, stable cardiopulmonary functions, a brain magnetic resonance imaging (MRI) scan within 2 weeks of enrollment demonstrating a measurable (≥10 mm diameter on T2W images), solitary intracranial lesion with imaging characteristics consistent with a meningioma, glioma, choroid plexus tumor, or nerve sheath tumor [1]. Dogs that had received prior therapy for the intracranial tumor were eligible if they met the following criteria: (1) ≥4 weeks elapsed since surgery or cytotoxic chemotherapy administration, (2) ≥4 months since radiation or immunotherapy treatments had been administered, and (3) progressive disease (≥40% increase in total T2 tumor volume [TTV] compared to MRI tumor nadir) present on pre-enrollment MRI [20]. Hepatic, renal, and bone marrow functional requirements were as follows: liver function tests ≤ 3 times the high-end of normal canine references ranges, creatinine ≤1.8 mg/dL, hematocrit ≥ 33%, segmented neutrophils ≥ 1500/mm^3^, and platelet count > 100,000/mm^3^. Dogs enrolled in the Phase II arm of the trial were also required to have a histologically confirmed brain tumor that demonstrated immunohistochemical reactivity to uPAR, as described previously [12]. Immunoreactivity to uPAR was semiquantitatively assessed, with a total possible score of 1–6 assigned representing the sum of the average percentage of immunoreactive cells present (1: 0–10% cells; 2: 10–50% cells; 3: 50–100% cells) and the degree of immunostaining intensity (0: None; 1: Weak; 2: Moderate; 3: Strong) in 5 high power (400×) fields. 

All doses of rLAS-uPA were administered as an intravenous (IV) infusion diluted in 150 mL sterile 0.9% saline delivered over 2 h, and each dog enrolled at each dose level received 3 IV rLAS-uPA infusions 14 days apart (Figure 1; Table 1). All dogs underwent continuous electrocardiographic and rectal temperature monitoring for the duration of the infusion. The primary endpoint of the trial was to determine the safety and maximum tolerated dose (MTD) of IV-administered rLAS-uPA. Safety was assessed by serial clinical and laboratory evaluations of each canine subject performed for up to 42 days (Phase I) or 180 days (Phase II) following rLAS-uPA treatment (Table 1), although clinical follow-up was continued off-protocol according to current best practices for management of brain tumors in some dogs. Each visit included physical and neurological examinations, KPS, laboratory assessments (CBC, serum biochemistry profile, urinalysis, +/− brain MRI and cerebrospinal fluid [CSF] analysis), and adverse event recording. Adverse events were classified and graded according to the Veterinary Cooperative Oncology Group Common Terminology Criteria for Adverse Events [21]. Dose-limiting toxicities (DLT) were defined as the presence of any grade 3, 4, or 5 adverse events that were possibly, probably, or definitely attributed to the viral infusion. The MTD was considered the dose level below the maximally administered dose at which ≤1 of at least 6 treated dogs experienced DLT (Figure 1). Clear exacerbation or progression of pre-existing tumor-associated clinical signs were not considered to be DLT. Dog owners were also trained to check their dog’s rectal temperature once daily for the duration of the study, and record the temperature on a study calendar.

A secondary endpoint was to evaluate the anti-tumor effects of the viral infusions as determined using serial brain MRI. MRI examinations were performed under general anesthesia using a 1.5T superconducting MRI system (Philips Intera, Andover, MA, USA) and standardized image acquisition protocol at 42, 84, and 180 days after initiation of rLAS-uPA therapy (Table 1) [22]. Quantitative MRI assessments of tumor response were determined using the T2 total tumor volumetric (TTV) technique, as not all tumors were contrast-enhancing, and defined as follows: a complete response (CR) required complete disappearance of all T2/FLAIR and contrast-enhancing lesions; a partial response (PR) required a ≥65% reduction in the TTV; progressive disease (PD) required ≥40% increase in the TTV [20]. All other responses were considered stable disease (SD).

Additional secondary clinical endpoints were the serial evaluation of KPS and caregiver-reported quality of life (QOL) scores, which served as surrogate measures of toxicity and anti-tumor effects as determined by clinicians and dog owners, respectively. The QOL was assessed with a survey instrument (CanBrainQOL-24) validated for use in dogs with brain disease [23]. CanBrain QOL-24 survey scores can range from 24 to 120, with higher scores indicative of a worse quality of life [23]. In the Phase II arm of the trial, the 6-month progression-free survival fraction was also used as an indicator of the anti-tumor activity of the rLAS-uPA.

### 2.3. Canine Tissue Collection and Preservation for In Vitro Viral and Immunologic Analyses

The schedule of biological tissue collection is provided in Table 1. Blood samples were collected via jugular or saphenous venipuncture into Vacutainer PST, serum, and K2EDTA (Becton-Dickinson, Franklin Lakes, NJ, USA) tubes for the performance of the hematologic and serum biochemical analyses. Complete blood counts and serum biochemical analyses were performed at a commercial veterinary diagnostic laboratory accredited by the American Association of Veterinary Laboratory Diagnosticians using routine, automated methods (Virginia Tech Animal Laboratory Services, Blacksburg, VA, USA). A 1 mL aliquot of whole blood from K2EDTA was then transferred to RNeasy protect animal blood tubes (Qiagen), and additional 1 mL aliquots of plasma and serum were harvested and frozen at −80 °C for future immunological assays. Urine was obtained using cystocentesis and free-catch techniques and collected into sterile plastic specimen receptacles. Urine samples were aliquoted into 8 mL sterile falcon tubes (Becton-Dickinson), centrifuged at 1000× *g* for 10 min at 4 °C, and the urine pellet was collected and suspended in 1 mL Qiazol (Qiagen). Urine samples were stored at −80 °C. While dogs were under general anesthesia and following the completion of brain MRI procedures, CSF was collected from the cerebellomedullary cistern and assayed immediately after collection using standard methods [24].

In select dogs undergoing surgical tumor resection following completion of the trial or necropsy examination, tumor samples were collected and immersed in 10% formalin or RNALater (Sigma-Aldrich, St. Louis, MO, USA), or flash frozen and stored at −80 °C. 

### 2.4. Statistical Analyses

Descriptive statistics were calculated for all quantitative variables. Longitudinal mixed effect models were used to compare KPS, QOL, CSF analytes, and serum cytokines to determine influences of timing related to viral dose, and treatment by time interactions. Statistical analyses were performed using SAS software (SAS v9.4, Cary, NC, USA)

## 3. Results

### 3.1. Canine Study Subjects and Tumor Characteristics

Twenty dogs were enrolled in the study (Figure 1). Individual dog demographic and tumor characteristics are provided in Appendix A. Represented breeds included mixed breeds (*n* = 3), Boston terriers (*n* = 2), Boxers (*n* = 2), Golden retrievers (*n* = 2), and *n* = 1 each of American Staffordshire terrier, Beagle, Border terrier, Chow, Fox terrier, French bulldog, German shepherd, Irish setter, Jack Russell terrier, Rottweiler, and Spitz. The median age was 9 years (range, 5–12 years), and the median body weight was 21.5 kg (range, 7–42 kg). There were 11 neutered males and 9 spayed females. All dogs had clinical signs of neurologic dysfunction reflective of the neuroanatomic location of the tumor within the brain, and the median duration of clinical signs prior to enrollment was 79 days (range, 3–521 days). The median KPS score at enrollment was 80 (range, 70–90). Tumors treated in the study included presumed meningiomas based on MRI features (*n* = 4), grade I meningioma (*n* = 5), grade II meningioma (*n* = 3), cranial nerve tumors (*n* = 2 [schwannoma and malignant peripheral nerve sheath tumor]), high-grade oligodendroglioma (*n* = 2), low-grade oligodendroglioma (*n* = 1), high-grade astrocytoma (*n* = 1), choroid plexus papilloma (grade I; *n* = 1), and choroid plexus carcinoma (grade III; *n* = 1) [1]. The median TTV at enrollment was 2.73 cm^3^ (range, 1.09–8.43 cm^3^). Among the tumor types with multiple grades represented, uPAR immunoreactivity scores were greater in high-grade tumors (Figure 2; Appendix A).

### 3.2. rLAS-uPA Viral Treatment, Safety and Neuroimaging Assessments, and Secondary Clinical Endpoints in Canine Subjects

rLAS-uPA-eGP was fusogenic in uPAR expressing canine J3T and human U87 glioma cells (Figure 3), and the starting dose (dose cohort 1; Figure 1) of rLAS-uPA administered to dogs was derived from the median tissue culture infective dose (TCID_50_) obtained from J3T canine glioma cell line experiments. A total of 59 individual rLAS-uPA viral infusions, representing four dose-escalation cohorts, were delivered to the 20 dogs over the course of the trial. A protocol deviation, characterized by administration of only two out of three planned cycles of rLAS-uPA occurred in one dog (Dog 11), as the owner refused further viral treatment at any dose due to concerns for the development of further adverse events (AE). Eighteen out of twenty (90%) dogs (11/11 in Phase I and 7/9 in Phase II) completed the trial. Dog 13 withdrew to receive other therapy on day 86, and Dog 18 was euthanized on day 107 due to progressive disease. Data from 20 dogs were included in endpoint analyses.

#### 3.2.1. Adverse Events in Canine Subjects

In total, 18/20 (90%) dogs experienced AE during the trial, and at least one AE in each of these 18 dogs was attributable to the rLAS-uPA treatment (Table 2). Two dogs experienced DLT characterized by high fever, anorexia, and thrombocytopenia or cerebral edema (Figure 1; Table 2) at the 2 × 10^9^ pfu/mL dose of rLAS-uPA. The MTD of rLAS-uPA was determined to be 2 × 10^7^ pfu/mL based on the administration of this dose to 12 dogs, none of which experienced DLT. Common, expected AE associated with rLAS-uPA treatment included generalized shivering during viral infusions (14 dogs), diarrhea (8 dogs), low-grade fever (5 dogs), and hyporexia (3 dogs). Other AE (Table 2) observed were attributed to pre-existing disease, concurrent medications, and non-investigational procedures. 

#### 3.2.2. Neuroimaging Evaluation of Anti-Tumor Effects in Canine Subjects

All dogs in the Phase I arm of the trial (Dogs 1–11) had stable disease on day 42 MRI examinations. In the nine dogs in the Phase II arm of the trial (Dogs 12–20), partial responses were seen in 2/9 (Dogs 12 and 19; Figure 4), stable disease in 6/9 (Dogs 13–17 and 20), and progressive disease in 1/9 (Dog 18). Three dogs treated at the MTD with stable disease had ≥40% but ≤65% tumor volumetric reductions (Dogs 7, 17, and 20; Figure 4), and thus did not meet objective threshold criteria for partial tumor responses. Multifocal regions of T1W hypointense, T2W/FLAIR hyperintense, and non-enhancing intratumoral fluid/cysts were observed in post-treatment MRI scans from three dogs with meningiomas and stable disease and minimal longitudinal change in tumor volume (Dogs 1, 4, and 13; Figure 4).

#### 3.2.3. Secondary Clinical Endpoints in Canine Subjects

For dogs in the Phase II trial arm, the 6-month progression-free survival fraction was 88% (7/8 dogs). There were no statistically significant time-dependent changes of KPS from baseline, although dogs in the maximum administered dose group experienced a significant decrease in post-treatment KPS scores (median 50, range 40–60) compared to baseline values (median 75, range 70–80; *p* = 0.03), with declines in KPS occurring in association with the occurrence of DLT, and returning to baseline values after resolution of DLT. Seventeen dogs (85%) experienced no change in KPS score from the baseline value over the entire course of the study. 

There were no statistically significant time-dependent changes in QOL scores. There was a trend for caregiver-reported QOL scores (Figure 5) to worsen (i.e., higher score) compared to baseline values during viral treatment (day 14). Post-treatment QOL scores (days 42 and 180) remained static and similar to baseline values in Dogs 13–15 and 19, were improved (i.e., lower score) in Dogs 12, 17, and 20, and worsened in Dogs 16 and 18. 

Statistically significant changes in CSF total protein concentrations and total nucleated cell counts following rLAS-uPA treatment were not observed (Appendix A). Additional off-protocol clinical follow-up information for individual canine subjects is provided in Appendix A. 

### 3.3. rLAS-uPA Viral and Immunologic Assays

#### 3.3.1. Infectious Viral Recovery and Viral Neutralizing Antibodies

No infectious virus was detected in plasma, urine, or CSF from any dog at any time point. Neutralizing antibodies against rLAS-uPA were detectable by day 14 in all dogs (Figure 6), with neutralizing antibodies persisting for the duration of the study in the majority of dogs. 

#### 3.3.2. rLAS-uPA qRT-PCR in Brain Tumors

Brain tumor samples were obtained from six dogs (Dogs 3, 6, 13, 15, 18, and 20) after completing rLAS-uPA treatment. Tumors included grade I meningioma (*n* = 3), grade II meningioma (*n* = 1), high-grade oligodendroglioma (*n* = 1), and choroid plexus carcinoma (*n* = 1). Samples were obtained by surgical resection after dogs completed the trial (*n* = 3) or at necropsy on- or off-protocol (*n* = 3). The median time period elapsed from the completion of viral treatment to collection of tumor samples was 142 days (range 76–304 days). rLAS-uPA viral gene expression was detected in tumors from all six dogs (Figure 7).

#### 3.3.3. Serum Cytokine Analysis

Serum TNF-α concentrations were significantly elevated from baseline in all dosing cohorts on day 14, and in dose cohorts 1, 3, and 4 on day 28, with levels returning towards baseline values by day 42 in all dosing cohorts (Figure 8). In all dosing cohorts, serum IFN-α and TRAIL concentrations increased significantly by day 14 and remained higher than baseline values for the duration of the study. Within each dosing cohort, serum IFN-α and TRAIL concentrations were also significantly different from each other at each sampling time (Figure 8). 

## 4. Discussion

This study provides the first characterization of the clinical toxicities, anti-tumor effects, and host-virus immunological responses of systemically administered oncolytic rLAS-uPA Newcastle disease virus administered to immunocompetent dogs with naturally occurring brain tumors. The data demonstrate that recombinant rLAS-uPA was capable of infecting a variety of uPAR-expressing canine brain tumors, generally well tolerated in tumor-bearing dogs, resulted in objective, neuroradiological tumor responses in 2/20 (10%) of the dogs enrolled in the study, and was associated with the rapid development of anti-rLAS-uPA neutralizing antibodies.

In dogs treated with rLAS-uPA doses at and below the MTD, mild and transient AE, such as chills during the infusion, fever, diarrhea, and decreased appetite, were commonly observed, as would be expected from treatment with a viral therapeutic [9,10]. The DLT observed in two dogs in this trial included both more severe non-specific manifestations of viral infection such as fever and anorexia, as well as thrombocytopenia and cerebral edema. The thrombocytopenia that was observed may have resulted from infection of platelets or megakaryocytes, as has been previously demonstrated with some strains of Newcastle disease virus [25]. Platelets also abundantly express uPAR, which plays a crucial role in platelet adhesion and trafficking in response to injury and disease, which may have increased the likelihood of developing thrombocytopenia in response to our rLAS-uPA-targeted virus [26]. The possibility that the thrombocytopenia was an immune-mediated event cannot be excluded, as idiopathic thrombocytopenic purpura is a recognized potential complication of acute viral infections [27]. Cerebral edema has been documented as an AE of oncolytic viral therapy in humans with brain tumors and is postulated to be the result of a potent viral-induced local inflammatory response [28]. In this study, the thrombocytopenia and cerebral edema both resolved with corticosteroid treatment. Given the inherent neurotropic features of Newcastle disease virus, the low incidence of nervous system AE attributable to the viral treatment observed in this study is encouraging. In this trial, AE that were attributable to concurrent medication or pre-existing disease have also been commonly observed in other studies investigating conventional treatments of canine brain tumors [5]. 

Using quantitative RT-PCR, it was demonstrated that rLAS-uPA was able to persistently colonize a subset of uPAR-expressing canine intracranial tumors following intravenous administration, including meningiomas, oligodendrogliomas, and choroid plexus tumors, which represent the most common primary brain tumors in dogs [1,5]. Two dogs (10%; 2/20) in this study experienced durable MRI-derived volumetric tumor reductions meeting the criteria for partial responses. Three dogs with stable disease also experienced >40% reductions in tumor volumes that did not meet the criteria for partial responses [20], and post-treatment intratumoral cystic regions were also observed in three dogs with stable disease. The significance of these cystic areas is unknown, as they could represent treatment-induced change, such as cystic tumor necrosis, but may also be a natural evolutionary feature of canine meningiomas [1]. While statistically significant improvements in KPS or QOL scores in the dogs in this study were not documented, and the overall objective neuroradiological response rate was modest, it is notable that an additional 85% (17/20) of dogs had stable disease over the course of the trial. For dogs in the Phase II trial arm, the 3- and 6-month progression-free survival fractions were 100% (9/9) and 88% (7/8 dogs), respectively. In the absence of definitive surgical, radiotherapeutic, or combinatorial cytotoxic treatments, the prognosis for dogs with brain tumors is poor, with disease progression typically resulting in death or euthanasia within two months of diagnosis [1,5]. Thus, the results suggest that rLAS-uPA treatment may be associated with clinically relevant benefits in some dogs with brain tumors and mild to moderate neurological dysfunction by delaying tumor progression without substantially reducing tumor volume. Given the limited numbers of different tumor types and objective responders, no attempt was made to statistically evaluate potential associations between tumor histology or uPAR target expression and neuroradiological response rates, although both responders in this study had tumors with high uPAR immunoreactivity. 

The infectious virus was not recovered from bodily fluids from any dog at any time point in this study, and all dogs developed persistent neutralizing antibodies to rLAS-uPA by day 14 post-treatment. These data are promising from a biosafety standpoint with respect to viral shedding in veterinary hospital and pet-owner home environments, given the potential pathogenicity of the Newcastle disease virus to avian species [8,11]. Optimal oncolytic viral dosing strategies remain controversial, with the majority of contemporary virotherapeutic studies adopting a dosing interval paradigm of either (1) single shot or repetitive dose-intensive administration within a few-day period, or (2) temporally dispersed repetitive dosing every few weeks [29]. Although the temporally dispersed, repetitive approach used in this study resulted in a favorable adverse effect profile, the rapidly induced host immune response may have significantly and negatively decreased the anti-tumor effects of the rLAS-uPA therapy [7,8,9,29]. Future studies should consider the incorporation of different rLAS-uPA dosing intervals or employ other strategies to evade the viral-induced immune response. For example, the incorporation of canine complement regulatory proteins into the Newcastle disease viral envelope by growing the virus in host-specific cells that are transfected with or naturally express the CD46 or CD55 complement regulatory proteins may confer resistance of the virus to neutralization by complement, or genetically engineering the virus to express immunostimulatory cytokines [30].

The oncolytic activity of NOV may occur by direct and indirect effects, with direct mechanisms including syncytial formation in tumor cells resulting in apoptosis, autophagy, or necrosis, and viral infection-inducing cytokine-mediated recruitment of innate and adaptive immune effector cells that exert anti-tumor effects [7,8,31]. This trial provides in vivo data illustrating the ability of rLAS-uPA to infect a variety of canine intracranial tumors for which there exist human homologs, in vitro evidence of the fusogenicity and syncytia-forming ability of the rLAS-uPA in human and canine glioma cell lines, and also significant increases in serum IFN-α, TNF-α, and TRAIL cytokines in dogs following viral treatment. 

The IFN-α response observed in dogs was not unanticipated, as activation of the type I IFN system is a widely conserved innate host response to viral infections, including Newcastle Disease virus, in mammalian cells [8,10,11]. The robust type I IFN response observed in canine subjects may have paradoxical biological implications. Induction of type I interferons is widely known to suppress viral replication and thus hamper viral-mediated oncolysis [7,30]. However, many cancer cell lines have been shown to have defects in IFN signaling pathways, which confers resistance to IFN-induced growth inhibition. This blunted innate anti-viral immune response in tumor cells may be a major mechanism that contributes to selective replication of Newcastle disease virus in neoplastic cells [7,8,31]. Several strains of the Newcastle disease virus have also evolved protective mechanisms to evade cellular type I IFN responses, although these adaptations may be species-specific to avian cells [32]. Conversely, type I IFN responses are fundamental to initiating and amplifying anti-tumor immune responses through the recruitment of natural killer and antigen-presenting cells, upregulation of MHC molecules, expression of cellular adhesion molecules that enhance fusogenicity on tumor cells, and priming of antigen-specific T cells [7,31]. 

Secretion of TNF-α by macrophages and Th1 cells following infection with Newcastle disease virus is a major mediator of viral-induced oncolysis [7]. TNF-α promotes the production of TRAIL, which then binds to cell surface death receptors and initiates apoptosis through caspase signaling. TRAIL-mediated cellular apoptosis demonstrates a selectivity for tumor cells, and this inherent selectivity is further enhanced by the upregulation of TRAIL receptor expression on cancer cells following infection with Newcastle disease virus [8,31]. Thus, while the serum cytokine signature of increased TNF-α and TRAIL expression observed are indicative of effector cell indirect anti-tumor responses following rLAS-uPA treatment, increased systemic circulating concentrations of these cytokines may also be causes of or contributors to the AE that were observed in the trial. The common AE observed in this study, such as chills during the viral infusion, fever, diarrhea, and hyporexia, are all known to be mediated by TNF-α [7,31]. This study further reinforces that major persistent impediments to the development of safe and effective oncolytic viral therapeutics are better understanding and balancing of the pleiotropic pro-inflammatory and anti-tumor immunological effects induced by viral infection [7,8,31]. To further characterize the systemic and local microenvironmental immune landscape of rLAS-uPA-treated tumors, comparative transcriptomic analyses in pre- and post-treatment tumor samples and expanded multiplex cytokine assays should be considered in future study designs. 

## 5. Conclusions

The favorable adverse effect profile and objective anti-tumor efficacy signals obtained against multiple tumor types in this study warrant further investigation of systemic administration of rLAS-uPA for the treatment of brain tumors. Data derived from canine large animal models of spontaneous brain cancers may be useful to inform the design and conduct of human oncolytic virotherapy studies. Future trials using rLAS-uPA should consider alternative dosing or additional immunomodulating viral engineering strategies to mitigate the host anti-viral immunity that may limit anti-tumor responses.

## Figures and Tables

**Figure 1 cancers-16-00564-f001:**
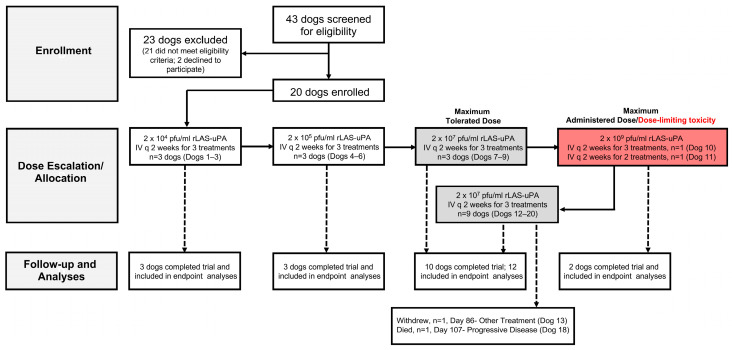
Phase I/II trial of urokinase plasminogen activator-targeted oncolytic Newcastle disease virus for canine intracranial tumors CONSORT diagram. The dosing cohort that experienced dose-limiting toxicities appears in red.

**Figure 2 cancers-16-00564-f002:**
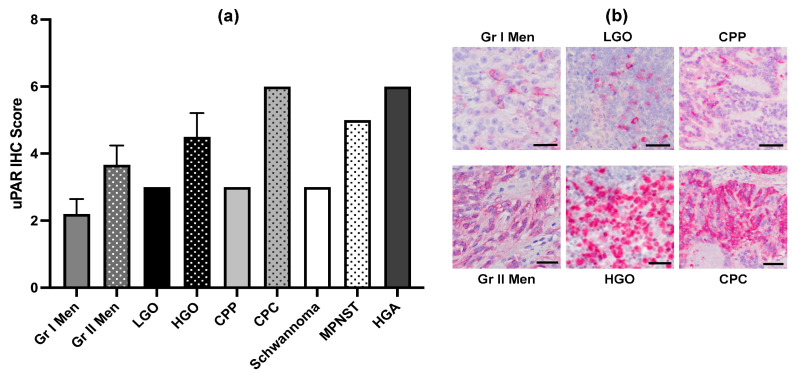
(**a**) uPAR immunohistochemistry (IHC) scores by tumor type and grade. For tumor types with multiple grades represented, higher-grade tumors are indicated by stippled bars. (**b**) Representative photomicrographs illustrating greater uPAR immunoreactivity, as indicated by red cytoplasmic staining of neoplastic cells, in higher tumor grades among tumors of the same type. Bar = 50 μm in all panels depicting fast-red chromogen with hematoxylin counterstain. Gr I Men—Grade I meningioma; Gr II Men—Grade II meningioma; LGO—low-grade oligodendroglioma; HGO—high-grade oligodendroglioma; CPP—choroid plexus papilloma (Grade I); CPC—choroid plexus carcinoma (Grade III); MPNST—malignant peripheral nerve sheath tumor; HGA—high-grade astrocytoma.

**Figure 3 cancers-16-00564-f003:**
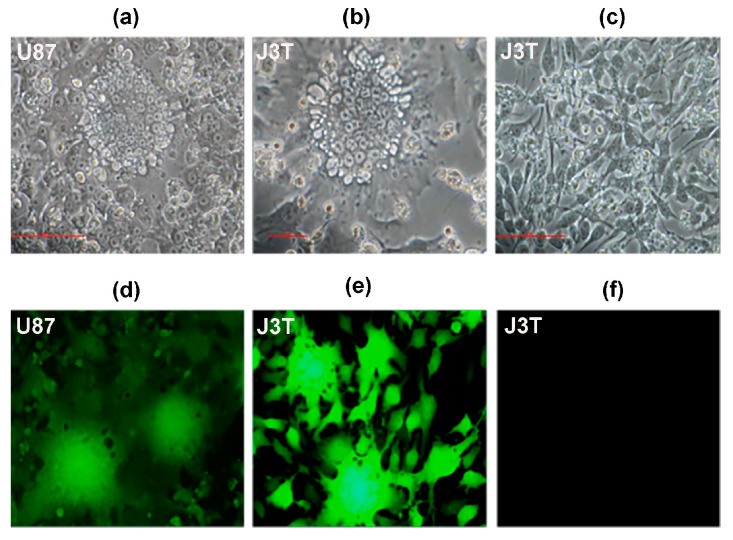
Tropism-modified and protease-activated rLAS-uPA is fusogenic in uPAR expressing canine and human glioma cells. Syncytia formation is present in human U87 and canine J3T glioma lines infected with rLAS-uPA-eGFP (**a**,**b**,**d**,**e**), but not those treated with PBS vehicle (**c**,**f**). Magnifcation of 200× for all panels.

**Figure 4 cancers-16-00564-f004:**
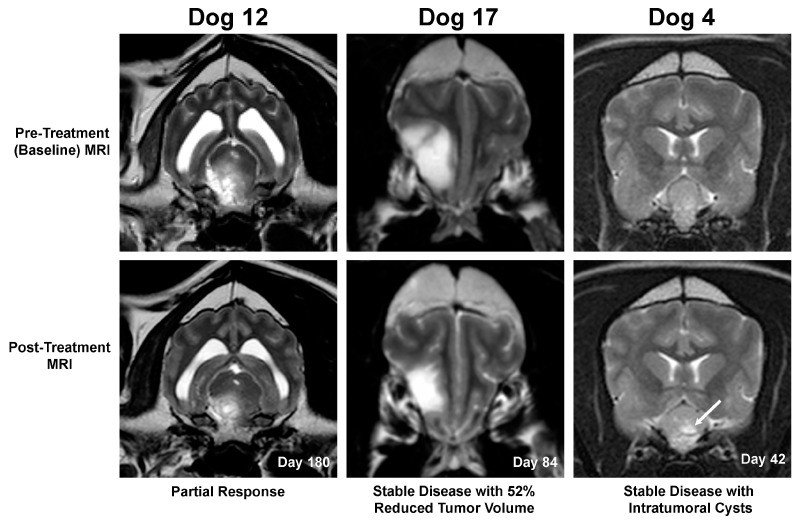
MRI-based objective characterization of tumor responses. Left panels, Dog 12—malignant peripheral nerve sheath tumor, illustrating partial tumor response. Middle panels, Dog 17—low-grade oligodendroglioma, stable disease associated with substantial reduction in tumor size. Right panels, Dog 4—parasellar meningioma, demonstrating the presence of areas of T2 hyperintense cystic change (arrow) in the tumor post-treatment.

**Figure 5 cancers-16-00564-f005:**
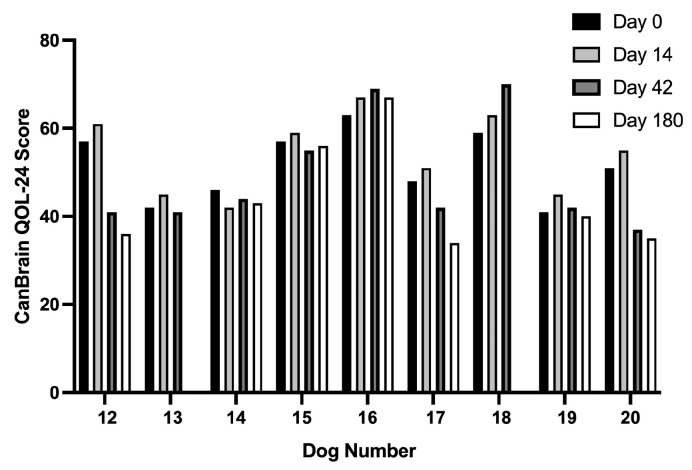
Quality of life (QOL) scores over the study period for dogs allocated to the Phase II trial arm. Higher CanBrain QOL-24 scores are associated with progressively more severe and detrimental physical and behavioral health impacts on the dog and overall worse QOL, as perceived by the pet caregiver [23].

**Figure 6 cancers-16-00564-f006:**
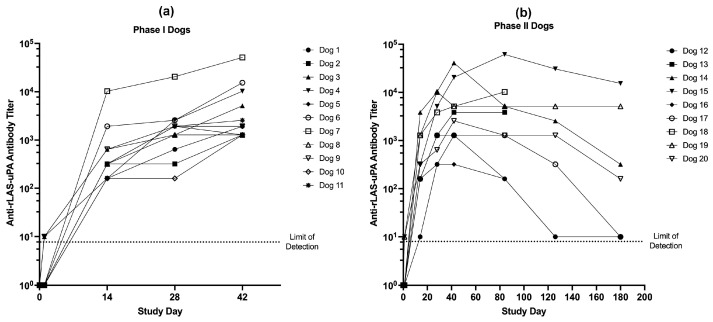
Anti-rLAS-uPA viral neutralizing titers in (**a**) Phase I and (**b**) Phase II canine subjects. All dogs rapidly developed neutralizing antibody titers that persisted for the duration of the trial.

**Figure 7 cancers-16-00564-f007:**
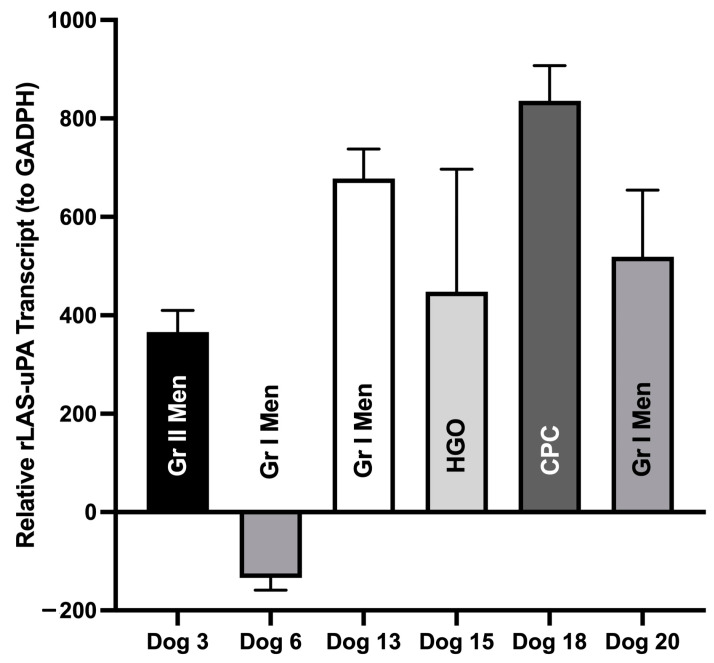
Quantitative TaqMan RT-PCR evaluating mRNA expression of rLAS-uPA in canine primary brain tumors following oncolytic viral therapy. rLAS-uPA transcripts were detected in all tumors. Gr I Men—Grade I meningioma; Gr II Men—Grade II meningioma; HGO—high-grade oligodendroglioma; CPC—choroid plexus carcinoma.

**Figure 8 cancers-16-00564-f008:**
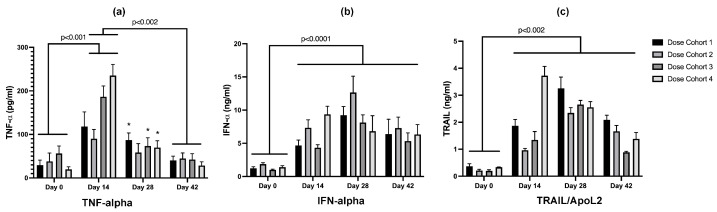
Longitudinal serum TNF-α (**a**), IFN-α (**b**), and TRAIL/ApoL2 (**c**) cytokine concentrations in rLAS-UPA dosing cohorts. At day 14, each cytokine in all dosing cohorts was increased significantly over baseline (day 0), with IFN-α, and TRAIL/ApoL2 remaining persistently and significantly elevated through study day 42. (*) Cytokine concentration different from all other time points within same dosing cohort.

**Table 1 cancers-16-00564-t001:** Phase I/II trial of urokinase plasminogen activator-targeted oncolytic Newcastle disease virus for canine intracranial tumors workflow and procedures.

Monitoring Parameter	Methods	Time Point (Trial Day)						
		−14 to 0	1	14	28	42	84	180
Owner	Observation		✓ Daily Days 1–42				X Daily Days 43–180	
	Rectal temperature							
	Adverse events							
		−14 to 0	1	14	28	42	84	180
Clinical	Physical/Neurological Exams	✓	✓	✓	✓	✓	X	X
	Karnofsky Performance Score	✓	✓	✓	✓	✓	X	X
	Adverse events	✓	✓	✓	✓	✓	X	X
Laboratory	CBC, Chemistry, Urinalysis	✓	✓	✓	✓	✓	X	X
	Cerebrospinal fluid	✓				✓	X	X
Viral Shedding	Blood and urine infectious viral recovery	†		†	†	†		
	Cerebrospinal fluid infectious viral recovery					†		
Viral Immune Response	Anti-NDV antibodies; plaque neutralization assay	✓	✓	✓	✓	✓	X	X
	Serum TNF-α, IFN-α, TRAIL; ELISA	†		†	†	†		
Tumor Response	Brain MRI	✓				✓	X	X
Quality of Life Assessment	CanBrain-24 QOL Caregiver Survey	X		X		X		X
Intervention	rLAS-uPA NDV IV Infusion		✓	✓	✓			

†—in Phase I trial arm only; ✓—in Phases I and II; X—in Phase II trial arm only; CBC—complete blood count; ELISA—Enzyme linked immunoassay; MRI—magnetic resonance imaging; NDV—Newcastle disease virus; QOL—quality of life.

**Table 2 cancers-16-00564-t002:** Summary of Adverse Events [21].

Adverse Events Attributable to rLAS-uPA Treatment			
Adverse Event	Category	Grade	Canine Subject (*n*, Dog Identification)
Fever	Constitutional	3/DLT	*n* = 2, Dog 10 ^^^, Dog 11 ^◊^
Anorexia	Gastrointestinal	3/DLT	*n* = 2, Dog 10 ^^^, Dog 11 ^◊^
Thrombocytopenia	Blood/Bone Marrow	4/DLT	*n* = 1, Dog 10 ^^^
Cerebral edema	Neurology	3/DLT	*n* = 1, Dog 11 ^◊^
Shivering/tremor	Administration Reaction	1	*n* = 14, Dogs 1 ^•^, 2 ^•^, 3 ^•^, 4 ^•^, 5 *, 7 ^•^, 9 *, 10 *, 11 ^◊^, 12 *, 13 ^•^, 15 ^•^, 17 ^•^, 18 *, 20 ^•^
Diarrhea	Gastrointestinal	1	*n* = 6, Dogs 1 *, 2 ^^^, 7 *, 15 *, 18 *, 20 ^•^
Diarrhea	Gastrointestinal	2	*n* = 2, Dogs 12 ^^^, 14 *
Fever	Constitutional	1	*n* = 4, Dogs 7 *, 9 *, 17 *, 19 *
Fever	Constitutional	2	*n* = 1, Dog 1 *
Appetite, altered	Gastrointestinal	1	*n* = 3, Dogs 10 *, 14 ^•^, 17 *
Conjunctivitis	Ocular	2	*n* = 1, Dog 14 *
Adverse Events Attributable to Concomitant Medication			
High alanine aminotransferase	Metabolic	1	*n* = 4, Dogs 1, 2, 15, 20 (prednisone, phenobarbital)
High alkaline phosphatase	Metabolic	1	*n* = 6, Dogs 1, 2, 3, 5, 13, 20 (prednisone, phenobarbital)
High alkaline phosphatase	Metabolic	2	*n* = 1, Dog 15 (prednisone, phenobarbital)
Adverse Events Attributable to Non-Investigational Protocol Procedures			
Cystitis/hematuria	Genitourinary	2	*n* = 1, Dog 15 (cystocentesis)
Hematoma	Hemorrhage/Bleeding	1	*n* = 1, Dog 7 (jugular venipuncture)
Adverse Events Attributable to Pre-existing Disease			
Seizures	Neurology	2	*n* = 8, Dogs 2, 3, 7, 8, 13, 14, 15, 19 (structural epilepsy)
Seizures	Neurology	4	*n* = 1, Dog 1 (structural epilepsy)
Depressed level of consciousness	Neurology	4	*n* = 1, Dog 18 (tumor progression)
Hydrocephalus	Neurology	3	*n* = 1, Dog 18 (tumor progression)

^•^ Adverse event occurred in association with each cycle of rLAS-uPA; * Adverse event occurred during/after 1st cycle of rLAS-uPA; ^◊^ Adverse event occurred during/after 2nd cycle of rLAS-uPA; ^^^ Adverse event occurred during/after 3rd cycle of rLAS-uPA.

## Data Availability

Data are contained within the article.

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
