# Peer review of "Phase I/II Trial of Urokinase Plasminogen Activator-Targeted Oncolytic Newcastle Disease Virus for Canine Intracranial Tumors"

_cancers, 2024, doi:10.3390/cancers16030564_

Round 1

Reviewer 1 Report

Comments and Suggestions for Authors

Dear Authors, the paper is very interesting and is prodromal for further investigations on this topic. By increasing the number of cases it will be possible to conduct an accurate statistical analysis study, not limited only to the descriptive one present in the article.

My only request, aimed at improving the scientific quality of the article, is to replace the sentences written in a personal form with impersonal verbal sentences.

Congratulations for the high scientific content of the manuscript.

Comments on the Quality of English Language

The authors, aimed at improving the scientific quality of the article, should to replace the sentences written in a personal form with impersonal verbal phrases.

Author Response

Dear Reviewer,

Thank you for your review of our manuscript.  In response to your concerns, we have replaced all uses of personal verbage in the revised paper, with changes highlighted in yellow

Reviewer 2 Report

Comments and Suggestions for Authors

The article entitled “Phase I/II trial of urokinase plasminogen activator targeted 2 oncolytic Newcastle disease virus for canine intracranial 3 tumors” is a very interesting research topic, providing valuable information on a clinical trial developed to use immunotherapy for treating dogs with brain tumors. The results are interesting and mainly focused on the safety of treatment and some dogs with clinical benefits. The article is well-written, and the authors provided the necessary methodology to allow data replication. I have only a few comments on the manuscript improvement. In lines, 124-125 authors described how the product was administrated.

1.     Authors should provide a subheading, specific to the rLAS-uPA w preparation and administration. The protocol used is separated into different parts of the text and makes it difficult to understand what was done in a chronological sequence. Then only later in the text, the authors described rLAS-uPA. Authors should prepare first and for then, administrate. Please, check this chronological order.

2.     In the rLAS-uPA preparation, it is not clear why each step is necessary and how each step is related to the final product. The authors could explain a little bit better. Why do we need cell lines? Please, provide a small sentence explaining the relation of the subheading with rLAS-uPA preparation.

3.     Seems that there is an unnecessary space at the beginning of some phrases. Please, check.  

4.     Figure 1. Please, increase quality.

5.     Figure 4, dog 4, the image is not very clear, seems that a little contrast was captured. Authors could include an arrow in the image to allow identification of the tumor area.

Author Response

Dear Reviewer,

Thank you for your thoughtful review and comments.  In response to your suggestions, we have made the following alterations to the revised manuscript: 1) we have rearranged the methods section such that the in vitro viral preparaton and analytical methodologies precede the description of the in vivo clinical trial methods and workflow, hopefully providing the chronological clarity you requested 2) we have expanded/clarification on the specific uses of the cell lines in the methods, 3) and made the requested revisions to Figures 1 and 4.  The spacing issues you identified may be a function of the journal's manuscript submission template, as we were unable to completely rectify these.  Perhaps they can be modified at the editorial level.